# Socioeconomic differentials in hypertension based on JNC7 and ACC/AHA 2017 guidelines mediated by body mass index: Evidence from Nepal demographic and health survey

Juwel Rana[1,2,3]*, Zobayer Ahmmad[4], Kanchan Kumar Sen[5], Sanjeev Bista[6], Rakibul M. Islam[7]

1 Research and Innovation Department, South Asia Institute for Social Transformation (SAIST), Dhaka, Bangladesh, 2 Department of Public Health, North South University, Dhaka, Bangladesh, 3 Department of Biostatistics and Epidemiology, University of Massachusetts Amherst, Amherst, Massachusetts, United States of America, 4 Department of Sociology, University of Utah, Salt Lake City, Utah, United States of America, 5 Department of Statistics, University of Dhaka, Dhaka, Bangladesh, 6 Advanced Biostatistics and Epidemiology, Ecole des Hautes Etudes en Sante Publique, Rennes, France, 7 Department of Epidemiology and Preventive Medicine, Monash University, Melbourne, Australia

* juwelranasoc@gmail.com

**Data Availability Statement:** All data files are available from the DHS program database: https://

## Abstract

### Background

Unlike developed countries, higher socioeconomic status (SES-education, and wealth) is associated with hypertension in low and middle-income countries (LMICs) with limited evidence. We examined the associations between SES and hypertension in Nepal and the extent to which these associations vary by sex and urbanity. The body mass index (BMI) was examined as a secondary outcome and assessed as a potential mediator.

### Materials and methods

We analyzed the latest Nepal Demographic and Health Survey data (N = 13,436) collected between June 2016 and January 2017, using a multistage stratified sampling technique. Participants aged 15 years or older from selected households were interviewed with an overall response rate of 97%. Primary outcomes were hypertension and normal blood pressure defined by the widely used Seventh Report of the Joint National Committee (JNC7) and the American College of Cardiology/American Heart Association (ACC/AHA) 2017.

### Results

The prevalence of hypertension was higher in Nepalese men than women. The likelihood of being hypertensive was significantly higher in the higher education group compared with the lowest or no education group for men (OR 1.89 95% CI: 1.36, 2.61) and for women (OR 1.20 95% CI: 0.79, 1.83). People in the richest group were more likely to be hypertensive compared with people in the poorest group for men (OR 1.66 95% CI: 1.26, 2.19) and for women (OR 1.60 95% CI: 1.20, 2.12). The associations between SES (education) and

dhsprogram.com/data/dataset/Nepal_Standard-DHS_2016.cfm?flag=0.

**Funding:** The author(s) received no specific funding for this work.

**Competing interests:** The authors have declared that no competing interests exist.

hypertension were partially modified by sex and fully modified by urbanity. BMI mediated these associations.

## Conclusions

The higher SES was positively associated with the higher likelihood of having hypertension in Nepal according to both JNC7 and ACC/AHA 2017 guidelines. These associations were mediated by BMI, which may help to explain broader socioeconomic differentials in cardio-vascular disease (CVD) and related risk factors, particularly in terms of education and wealth. Our study suggests that the mediating factor of BMI should be tackled to diminish the risk of CVD in people with higher SES in LMICs.

## Introduction

Hypertension is a growing public health problem in low and middle-income countries (LMICs) with concurrent risks of cardiovascular and kidney diseases [1, 2]. A review warned that although about three-quarters of people with hypertension (639 million people) live in LMICs, there is no improvement in awareness or control rates [1]. Hypertension is a major contributor to death and disability in South Asian countries, including Nepal with a low level of control and awareness [3–6]. The World Health Organization (WHO) implemented 'STEP-wise approach to surveillance' (STEPS) using nationally representative sample in 2008 and 2013 reported an increasing trend of prevalence of hypertension among 15–69 years Nepalese population ranging from 21.5% in 2008 to 26.0% in 2013 [7, 8]. Based on the recent Nepal Demographic and Health Survey (NDHS) 2016, Kibria and colleagues reported that the esti-mated prevalence of hypertension in Nepal using the widely used Seventh Report of the Joint National Committee (JNC7) guideline was 21.2%, and the corresponding prevalence was 44.2% when using a new hypertension guideline recommended by the American College of Cardiology/American Heart Association 2017 (ACC/AHA 2017) [9, 10]. This study demon-strated that the prevalence of hypertension increased to 23% when using new ACC/AHA guideline, with the highest increase in the richest and obese population [11].

Despite an increasing prevalence of hypertension in Nepal, research exploring complex interrelationship between socioeconomic status (SES), indicated by education levels and wealth quintiles, and hypertension is limited. Moreover, this association is complex, unlike developed countries, in LIMCs. For instance, the prevalence of hypertension is higher among low SES groups in developed countries, while it is substantially higher among high SES groups in LMICs [12–15].

The reasons for the high prevalence of hypertension in the low SES group in developed countries include higher smoking rates, higher body mass index (BMI), and lack of exercise compared with higher SES groups [16]. The opposite pattern is observed in LMICs, where a higher prevalence of these risk factors is observed in higher SES groups compared with low SES groups. A recent review found that people in higher SES groups in LMICs were less likely to be physically active and consume more fats, salt, and processed food than low SES groups [17]. Furthermore, studies also found that BMI is exponentially increasing in people in LIMCs, which are the key modifiable risk factors for hypertension [18–20]. Thus, we hypothe-sized that there would be positive associations between high SES and hypertension in Nepal, and the level of BMI will at least partially mediate these associations. The primary aim of this study was (i) to assess the associations between SES and hypertension in Nepal, and the extent

to which these associations vary by gender and urbanity; and (ii) to examine whether BMI attenuates the associations between SES and hypertension and, the extent to which BMI explain these associations. The secondary aim of this study was to examine associations of BMI with SES and the extent to which these associations vary by gender and urbanity.

## Materials and methods

### Data source

The study analyzed the nationally representative Nepal Demographic and Health Survey (NDHS) 2016 data, collected between June 2016 and January 2017. The Nepal Health Research Council and the ICF International institutional review board approved the NDHS 2016 survey protocol. The household head provided written informed consent before the interview. For the current study, we obtained approval to use the data from ICF in June 2018.

### Survey design and study populations

The updated version of the census frame of the National Population and Housing Census 2011 conducted by the Central Bureau of Statistics was used as the sampling frame for the NDHS 2016. The households of the NDHS 2016 were selected in two ways based on the urban/rural locations. Firstly, the two-stage stratified sampling process was used in rural areas where wards were selected in the first stage as a primary sampling unit (PSUs), and households were selected in the second stage. Secondly, three-stage stratified sampling was used in urban areas to select households where wards were selected in the first stage (PSUs), enumeration areas (EA) were selected from each PSU in the second stage and households were selected from EAs in the third stage. There were 14 sampling strata in the NDHS 2016, where wards were selected randomly from each stratum. A total of 383 wards were selected altogether, 184 from urban and 199 from rural areas. Finally, a total of 11,490 households (rural- 5,970 and urban-5,520) were selected for the NDHS 2016 [21]. Flowchart of the analytic sample selection process is given as a supplementary figure (S2 File).

The trained interviewers collected data visiting the households. The overall response rate was approximately 97%. Blood pressure (BP) was measured among 15,163 individuals with 6,394 men and 8,769 women aged 15 years and above. In our study, a total unweighted sample was 13,371 comprising men (5,535) and women (7,836), after excluding participants aged <18 years and discarding the missing and extreme values. The total weighted analytic sample was 13,436 participants (men 5,646 and women 7,790) aged 18 years and above. Details of the NDHS 2016, including survey design, sample size determination, and questionnaires, have been described elsewhere [21].

### Measures of outcomes: Blood pressure outcomes

Hypertension and normal blood pressure were considered as the outcome variables in the study defined by both JNC7, and ACC/AHA 2017 guidelines (Table 1) [9, 10]. Three measurements of blood pressure (systolic and diastolic blood pressures) were taken for each participant with an interval of 5 minutes between the measurements by UA-767F/FAC (A&D Medical) blood pressure monitor. Systolic blood pressure (SBP) and diastolic blood pressure (DBP) were defined by taking the average of three SBP and three DBP measurements, respectively. We used both 'measurement-only' and 'medical/clinical' definitions to generate independent binary outcomes for 'hypertension' and 'normal blood pressure' based on both guidelines. The 'measurement-only' definition was developed solely based on the cut-off points that accounted for the average of three SBP and three DBP measurements. The 'medical/clinical' definition

**Table 1. Definitions of blood pressure outcome used in the study.**

| Blood pressure outcomes | Measurement-only definitions | Medical definitions |
|---|---|---|
| Hypertension (JNC7)[†] | SBP ≥140mmHg or DBP ≥ 90 mmHg | Meet any of the following three criteria:<br>(1) SBP ≥ 140mmHg or DBP ≥ 90mmHg<br>(2) Doctor/nurse diagnosed high blood pressure<br>(3) Taking blood pressure-lowering medication |
| Hypertension (ACC/AHA 2017)[‡] | SBP ≥130mmHg or DBP ≥ 80 mmHg | Meet any of the following three criteria:<br>(1) SBP ≥ 130mmHg or DBP ≥ 80mmHg<br>(2) Doctor/nurse diagnosed high blood pressure<br>(3) Taking blood pressure-lowering medication |
| Normal blood pressure (JNC7 or ACC/AHA 2017) | SBP <120mmHg and DBP <80 mmHg | SBP ≤ 120mmHg and DBP ≤ 80 mmHg, no diagnosis of high blood pressure, and not taking blood pressure-lowering medication |

[†]JNC7 = The Seventh Report of the Joint National Committee

[‡]ACC/AHA 2017 = The 2017 American College of Cardiology/American Heart Association

accounted for 'measurement-only' definition plus medical diagnosis by a health professional as having high blood pressure and/or taking blood pressure-lowering medication (Table 1).

## Measures of exposure: Socioeconomic status

Three indicators such as education levels, wealth quintiles, and employment status are most commonly used in several studies to assess the SES of a participant [12, 15]. However, we omitted employment status from our assessment of SES and subsequent analyses due to a large number of missing values as the majority of the women in South Asia are not involved in the formal employment. The NDHS 2016 provided data for a derived wealth quintile using the principal component analysis taking scores of a household's durable and nondurable assets. Firstly, households are given scores using principal component analysis based on the number and kinds of consumer goods they own. Secondly, to get the wealth quintiles, the distribution of scores was divided into five equal sizes named as poorest, poorer, middle, richer, and richest. Education was an ordinal measure of self-reported levels of education, which was grouped into four different categories (no education/preschool, primary, secondary, and higher education) in the NDHS 2016. In our study, the SES measures were not indexed for two main reasons. Firstly, different indicators of SES tend to have different theoretical pathways to BMI and blood pressure outcomes. Secondly, SES indicators might be causally related to each other; and they build on each other according to the life course models [22].

## Body mass index

The BMI was used in the study as both continuous and categorical variables. We followed both the South-Asian specific and global definition of BMI.

## Statistical analysis

Our primary statistical analyses assessed the sex and urbanity stratified associations of educational levels and wealth quintiles with blood pressure outcomes using both the measurement-only and medical definitions. To characterize the shapes of the associations, we calculated sex and urbanity stratified adjusted odds ratios (ORs) and 95% confidence intervals (CIs) within each level of education or wealth quintiles by using the binary logistic regression models. We used a cut off of 10% change in the stratified analysis as well as tested interaction term to identify differences in hypertension by sex and urbanity.

We further tested whether BMI mediates the associations between SES and blood pressure outcomes. We employed the following two approaches for testing mediation effect of BMI: The first approach was the "reduction-in-estimate criterion," approach- a rule of thumb, which assessed whether the inclusion of mediator variable-BMI attenuated the associations or effects for the main predictors across nested models. Hence, we constructed two nested models stratified by sex, and coefficients were progressively adjusted for age, marital status, urbanity, and second-hand smoking in Models I(a), II(a), III(a). Coefficients were further adjusted for prior determined mediator-BMI in Models I(b), II(b), III(b) to observe changes in the coefficients of predictors. We considered that there is a mediation effect using a cut off of 10% reduction in the effect estimate (coefficients) after adjusting for mediator-BMI in the respective models.

The second step was the "indirect effect" approach, which formally examined the statistical significance of an indirect effect using the product of coefficients approach [23]. For assessing the indirect effect of BMI on these binary outcomes, we used the generalized structural equation modeling (GSEM) in Stata because this approach is commonly used and can detect which variables are continuous and which are binary. It requires information for each link in the proposed mediation process [Mediator Variable (MV) regressed on Independent Variable (IV) and Covariates (CV) and Dependent Variable (DV) regressed on MV, IV, and CV] [24, 25]. In supplemental analyses, we replicated the process for 5000 bootstraps for statistical significance, which provided substantially identical indirect effects along with standard errors and biased-corrected 95% CI for the indirect effect of SES [24].

Additionally, we examined the adjusted associations between SES and BMI as a continuous outcome. Moreover, the adjusted sex and urbanity stratified associations between SES and binary outcome-overweight/obese (using both global and South Asia-specific cut-offs for BMI) were assessed to observe differences in overweight/obesity by sex and urbanity.

For examining the associations between SES and hypertension, all potential confounders for each predictor were selected using prior knowledge and directed acyclic graphs (DAGs) to avoid the 'Table 2 fallacy' in a multivariable model and to observe unbiased total effect estimates for predictors [26, 27].

For the brevity, we have reported 'measurement-only definition' of hypertension/normal blood pressure, if not stated otherwise, especially when we assessed associations between hypertension/normal blood pressure and SES. However, similar analyses for 'medical definition' of hypertension have been provided as supplementary data (S1 File). Comparable analyses based on the new guideline of ACC/AHA 2017 have also been given as supplementary data. Two-sided P-values and 95% CIs are presented. The complex survey design effects were accounted in all performed analyses for reducing differences due to oversampling, variation in the probability of selection and non-response in the NDHS 2016. All analyses were performed using Stata 15 (StataCorp).

## Results

### General characteristics of study participants

Of 13,436 participants, 7,790 (58%) were women, and 5,645 (42%) were men, with a mean age of 40.7 (SE ±0.10) years (Table 2). More than half (61.1%) of the population lived in urban areas with no significant sex difference. About 40% of the population had no education, and men were more likely to be educated than women at each level of education (p <0.001). Men were also more likely to be wealthier than women were (p <0.0001).

A similar trend was found for employment status where men were about 24% higher in employment status than women were (p <0.001). Mean BMI was significantly higher among

**Table 2. Sample characteristics (weighted numbers and percentages unless stated otherwise).**

| Characteristics | Overall (n = 13 436) | Men (n = 5645) | Women (n = 7790) | p-value |
|---|---|---|---|---|
| **Mean Age (SE, standard error)** | 40.7 (0.1) | 42.59 (0.28) | 39.30 (0.19) | < 0.001 |
| **Marital Status (%)** | | | | |
| Unmarried | 1569 (11.7) | 872 (15.4) | 698 (9.0) | < 0.001 |
| Married | 11 867 (88.3) | 4 774 (84.6) | 7092 (91.0) | |
| **Education Levels (%)** | | | | |
| No education/preschool | 5498 (40.9) | 1474 (26.1) | 4024 (51.7) | < 0.001 |
| Primary | 2281 (17.0) | 1194 (21.2) | 1087 (14.0) | |
| Secondary | 3709 (27.6) | 1958 (34.7) | 1751 (22.5) | |
| Higher | 1947 (14.5) | 1020 (18.1) | 928 (11.9) | |
| **Employment Status (%)** | | | | |
| Unemployed | 2777 (30.6) | 557 (15.9) | 2,220 (40.0) | < 0.001 |
| Employed | 6287 (69.4) | 2956 (84.15) | 3331 (60.0) | |
| **Wealth Index (%)** | | | | |
| Poorest | 2405 (17.9) | 993 (17.6) | 1412 (18.1) | < 0.001 |
| Poorer | 2613 (19.5) | 1054 (18.7) | 1559 (20.0) | |
| Middle | 2693 (20.0) | 1091 (19.3) | 1603 (20.6) | |
| Richer | 2936 (21.9) | 1280 (22.8) | 1656 (21.3) | |
| Richest | 2787 (20.8) | 1228 (21.8) | 1559 (20.0) | |
| **Urbanity (%)** | | | | |
| Urban | 8205 (61.1) | 3475 (61.6) | 4729 (60.7) | 0.27 |
| Rural | 5231 (38.9) | 2171 (38.6) | 3061 (39.3) | |
| **Region (%)** | | | | |
| Mountain | 859 (6.4) | 367 (6.5) | 491 (6.3) | 0.60 |
| Hill | 5922 (44.1) | 2468 (43.7) | 3454 (44.3) | |
| Terai | 6655 (49.5) | 2811 (49.8) | 3844 (49.4) | |
| **Established Risk Factors of Hypertension** | | | | |
| Mean Systolic Blood Pressure (SE) | 117.7 (0.2) | 122.02 (0.43) | 114.57 (0.38) | < 0.001 |
| Mean Diastolic Blood Pressure (SE) | 78.3 (0.1) | 79.89 (0.32) | 77.17 (0.26) | < 0.001 |
| High Blood Pressure (Told by doctor, %) | 1670 (12.4) | 763 (13.52) | 907 (11.64) | 0.004 |
| Medication for Blood Pressure (%) | 578 (4.3) | 260 (4.61) | 318 (4.08) | 0.25 |
| Mean Body Mass Index (SE) | 22.1 (0.0) | 21.08 (0.72) | 22.28 (0.10) | < 0.001 |
| Exposure to Secondhand Smoking (%) | 6308 (47.0) | 2718 (48.2) | 3589 (46.08) | 0.003 |
| Consumption of Caffeine (%) | 1058 (8.0) | 581 (10.3) | 477 (6.12) | < 0.001 |

women (22.28 vs. 21.08; $p < 0.001$) compared with men. Men were more likely to be exposed to secondhand smoking (p < 0.003) compared with women.

## Prevalence of hypertension by sex and urbanity

Women were having lower prevalence of hypertension compared with men for both measured (16.0%, 95% CI: 14.8, 17.3 vs. 22.8%, 95% CI: 21.2, 24.5) and medical hypertension (21.7%, 95% CI: 20.4, 23.0 vs. 29.1%, 95% CI: 27.4, 30.8) and the differences were significant statistically in both measurements ($p < 0.001$) (Table 3). People living in urban areas were having higher prevalence of hypertension compared with people living in rural areas for both measured (19.5%, 95% CI: 18.7, 20.4 vs. 17.9%; 95% CI: 16.9, 19.0) and medical (26.2%, 95% CI: 25.2, 27.1 vs. 22.7%; 95% CI: 21.6, 23.8) hypertension and the differences were significant statistically ($p < 0.001$) only for medical hypertension. Comparable trends were observed for both

**Table 3. Prevalence of hypertension by sex and urbanity in Nepal.**

| Classification of Blood Pressure | Overall n = 13 436 (%) [95% CI] | Male n = 5645 (%) [95% CI] | Female n = 7790 (%) [95% CI] | p value | Urban n = 8,205 (%) [95% CI] | Rural n = 5,231 (%) [95% CI] | p value |
|---|---|---|---|---|---|---|---|
| **JNC7 Guideline** | | | | | | | |
| Hypertension (measured) | 2538 (18.9) [17.7, 20.1] | 1289 (22.8) [21.2, 24.5] | 1249 (16.0) [14.8, 17.3] | < 0.001 | 1600 (19.5) [18.7, 20.4] | 938 (17.9) [16.9, 19.0] | 0.22 |
| Hypertension (medical) | 3333 (24.8) [23.6, 26.0] | 1645 (29.1) [27.4, 30.8] | 1688 (21.7) [20.4, 23.0] | < 0.001 | 2147 (26.2) [25.2, 27.1] | 1186 (22.7) [21.6, 23.8] | 0.007 |
| Normal Blood Pressure (measured) | 7233 (53.8) [52.1, 55.6] | 2581 (45.7) [43.5–48.0] | 4652 (59.7) [57.9, 61.5] | < 0.001 | 4340 (52.9) [51.8, 54.0] | 2893 (55.3) [54.0, 56.7] | 0.22 |
| Normal Blood Pressure (medical) | 6888 (51.3) [49.7, 52.9] | 2449 (43.4) [41.2, 45.6] | 4439 (57.0) [55.3, 58.7] | < 0.001 | 4113 (50.1) [49.1, 51.2] | 2775 (53.1) [51.7, 54.4] | 0.11 |
| **ACC/AHA 2017 Guideline** | | | | | | | |
| Hypertension (measured) | 5728 (42.6) [40.9, 44.4] | 2772 (49.1) [47.8, 50.4] | 2956 (38.0) [36.9, 39.0] | < 0.001 | 3582 (43.7) [42.6, 44.7] | 2146 (41.0) [39.7, 42.4] | 0.18 |
| Hypertension (medical) | 6136 (45.7) [44.1, 47.3] | 2950 (52.3) [51.0, 53.6] | 3186 (40.9) [39.8, 42.0] | < 0.001 | 3848 (46.9) [45.8, 48.0] | 2288 (43.7) [42.4, 45.1] | 0.08 |
| Normal Blood Pressure (measured) | 7093 (52.8) [52.0, 53.6] | 2524 (44.7) [43.4, 46.0] | 4569 (58.7) [57.6, 59.7] | < 0.001 | 4251 (51.8) [50.7, 52.9] | 2842 (54.3) [53.0, 55.7] | 0.20 |
| Normal Blood Pressure (medical) | 6763 (50.3) [49.5, 51.2] | 2397 (42.5) [41.2, 43.8] | 4365 (56.0) [54.9, 57.1] | < 0.001 | 4034 (49.2) [48.1, 50.3] | 2729 (52.2) [50.8, 53.5] | 0.10 |

measurements in normal blood pressure (p <0.001). According to the new ACC/AHA 2017 guideline, there was an overall 21% increase in the prevalence of hypertension, with the highest increase in the male population (23%). Similar trends of sex differences were observed in hypertension (p <0.001) by both guidelines; however, significant urban-rural differences (p >0.05) were not observed.

## Socioeconomic status and hypertension by sex and urbanity

Figs 1 and 2 explained the odds of blood pressure outcomes by education and wealth quintiles. The likelihood of being hypertensive (measured) was significantly higher in the higher education group compared with the lowest or no education group for men (OR 1.89 95% CI: 1.36, 2.61) and for women (OR 1.20 95% CI: 0.79, 1.83). People in the richest group were more likely to be hypertensive (measured) compared with people in the poorest group for men (OR 1.66 95% CI: 1.26, 2.19) and women (OR 1.60 95% CI: 1.20, 2.12). The overall associations between SES and hypertension were positive and statistically significant, modified by urbanity. However, the association between education and hypertension, not wealth and hypertension, was modified by gender (S1 File). Similar trends and associations between hypertension and SES were observed for ACC/AHA 2017 guidelines, and the effect of SES was modified by gender and urbanity (S2 File). Similarly, people with higher SES were less likely to have normal blood pressure compared with people in low SES (Figs 1 and 2, S1 File).

## Mediation effect of BMI on SES and hypertension

Table 4 shows a reduction in estimates of SES after adjusting for mediator variable-BMI in the logistic regression models. At least a 10% change in the regression coefficients due to adjusting for mediator indicates its mediating effect. For the levels of education, the adjusted odds of hypertension (measured) significantly decreased, at least 10% throughout the models, and particularly BMI attenuated the association and level of significance for each primary, secondary

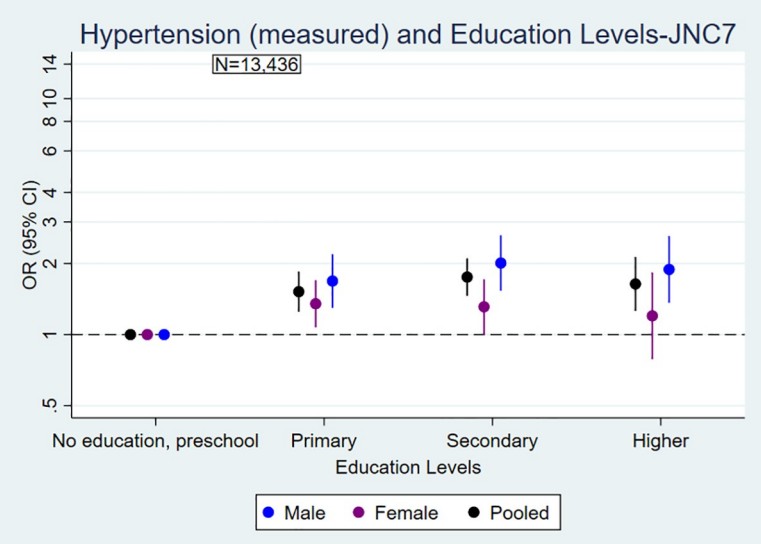

**Fig 1.** Association of (a) hypertension and (b) normal blood pressure (measured) with education levels by sex in Nepal. a) Hypertension and Education Levels b) Normal Blood Pressure and Education Levels. Odds ratios are adjusted for age, urbanity and marital status, and stratified by sex. Measurement-only outcomes are defined based on cut-off points: hypertension: SBP ≥140mmHg or DBP ≥90mmHg; normal blood pressure: SBP ≤ 120mmHg and DBP ≤ 80 mmHg.

and higher education category with hypertension (measured) in the models I, II and III. Table 4 also suggests that further adjustment for mediator-BMI in models reduced the effect size and level of significance in wealth quintiles and hypertension (measured). In other words, the further inclusion of BMI in the models has reduced the regression coefficients of

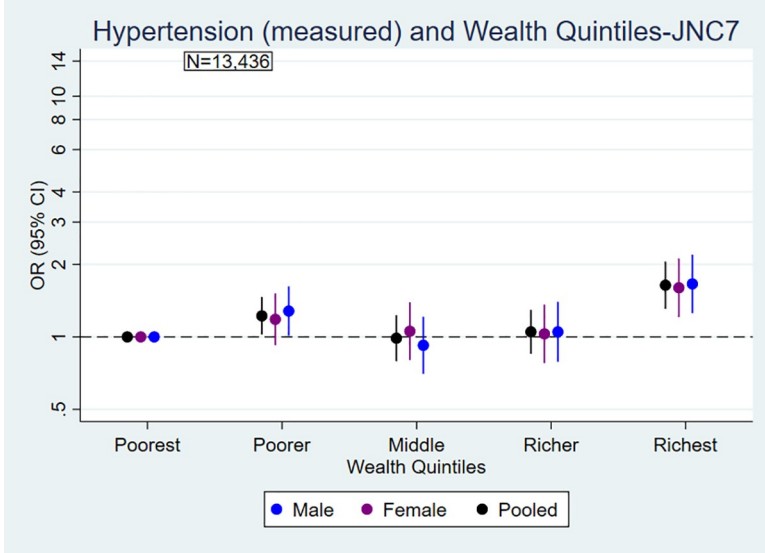

**Fig 2.** Association of (a) hypertension and (b) normal blood pressure (measured) with wealth quintiles by sex in Nepal. a) Hypertension and Wealth Quintiles b) Normal Blood Pressure and Wealth Quintiles. Odds ratios are adjusted for age, urbanity and marital status, and stratified by sex. Measurement-only outcomes are defined based on cut-off points: hypertension: SBP ≥140mmHg or DBP ≥90mmHg; normal blood pressure: SBP ≤ 120mmHg and DBP ≤ 80 mmHg.

**Table 4. Mediation effect (by 10% change in coefficients after adjusting for the mediator) of BMI on SES and hypertension by sex in Nepal.**

| Predictor | Model I- Overall (n = 13,436) | | Model II-Men (n = 5,646) | | Model III-Women (n = 7,790) | |
|---|---|---|---|---|---|---|
| | [a]Regression Coefficients without mediator (95% CI) | [b]Mediator Adjusted Regression Coefficients (95% CI) | [a]Regression Coefficients without mediator (95% CI) | [b]Mediator Adjusted Regression Coefficients (95% CI) | [a]Regression Coefficients without mediator (95% CI) | [b]Mediator Adjusted Regression Coefficients (95% CI) |
| **Hypertension (measured) by Education Levels (Ref. No education/preschool)** | | | | | | |
| Primary | 0.33 (0.15, 0.52)*** | 0.19 (-0.01, 0.38) | 0.53 (0.27, 0.79) *** | 0.40 (0.13,0.66) ** | 0.29 (0.06,0.51) ** | 0.11 (-0.13, 0.35) |
| Secondary | 0.46 (0.27, 0.65)*** | 0.24 (0.05, 0.43)* | 0.73 (0.46, 1.00) *** | 0.48 (0.19,0.77) *** | 0.24 (-0.02.0.51) | 0.03 (-0.23,0.30) |
| Higher | 0.39 (0.14, 0.65) ** | 0.11 (-0.16, 0.38) | 0.69 (0.36, 1.01) *** | 0.34 (-0.01, 0.69) * | 0.13 (-0.28,0.55) | -0.10 (-0.53,0.33) |
| **Hypertension (measured) by Wealth Quintiles (Ref. Poorest)** | | | | | | |
| Poorer | 0.20 (0.02,0.39) * | 0.17 (-0.01, 0.35) | 0.27 (0.03,0.50) * | 0.22 (-0.02,0.45) | 0.15 (-0.10,0.40) | 0.12 (-0.13,0.37) |
| Middle | -0.01 (-0.24,0.21) | -0.07 (-0.29,0.15) | -0.06 (-0.33,0.22) | -0.13 (-0.41,0.15) | 0.03 (-0.25,0.31) | -0,01 (-0.29,0.26) |
| Richer | 0.04 (-0.18,0.26) | -0.14 (-0.35,0.08) | 0.09 (-0.21,0.38) | -0.09 (-0.39,0.20) | -0.01 (-0.29,0.28) | -0.20 (-0.48,0.08) |
| Richest | 0.49 (0.26, 0.73) *** | 0.04 (-0.20,0.29) | 0.57 (0.28,0.86) *** | 0.16 (-0.17,0.49) | 0.42 (0.13,0.70) ** | -0.10 (-0.41,0.21) |

aCoefficients adjusted for age, sex, marital status, urbanity, and second-hand smoking; bCoefficients further adjusted for mediator-BMI. Regression coefficients; 95% confidence intervals in brackets

* p<0.05

** p<0.01

*** p<0.001.

hypertension-at least 10% for wealth quintiles and reduced statistical significance (Table 4: Model I(a) vs. Model I(b); Model II(a) vs. Model II(b); Model III(a) vs. Model III (b)). BMI, therefore, may play a mediating role in the associations between SES and hypertension (measured) for both men and women. Similar analyses were also performed for hypertension (medical) [S1 File].

## Mediation Analysis: Body Mass Index

The average BMI, according to the global cut-offs of BMI, was about 22, which indicates about 18% of the respondents were obese/overweight. However, the prevalence of obesity/overweight, according to the South Asia-specific cut-offs of BMI, was about 37%. The likelihood of being overweight/obese increased with an increasing level of SES, which also modified by sex and urbanity (S1 and S2 Files). Hence, we formally tested the mediation effect of BMI on hypertension (measured) and SES as well as presented the path coefficients (95% CI), and indirect effects of SES through BMI with bias-corrected 95% CI (Fig 3). The indirect effect of education on hypertension through BMI was statistically significant (Coef. 0.48; 95% bias-corrected CI: 0.41, 0.56). The total direct effect of education levels was Coef. 0.82 (95% CI: 0.48, 1.17). Thus, we may interpret that BMI mediated about 37% of the effect of education on hypertension. Similarly, the indirect effect of wealth quintiles on hypertension through BMI (Coef. 0.71; 95% bias-corrected CI: 0.61, 0.82) was significant. The direct effect was Coef. 0.09 (95% CI: -0.41, 0.58), and BMI mediated 89% of the total effect of wealth quintiles. BMI played a similar mediating role in the associations between SES and hypertension by ACC/AHA 2017 guidelines (S2 File).

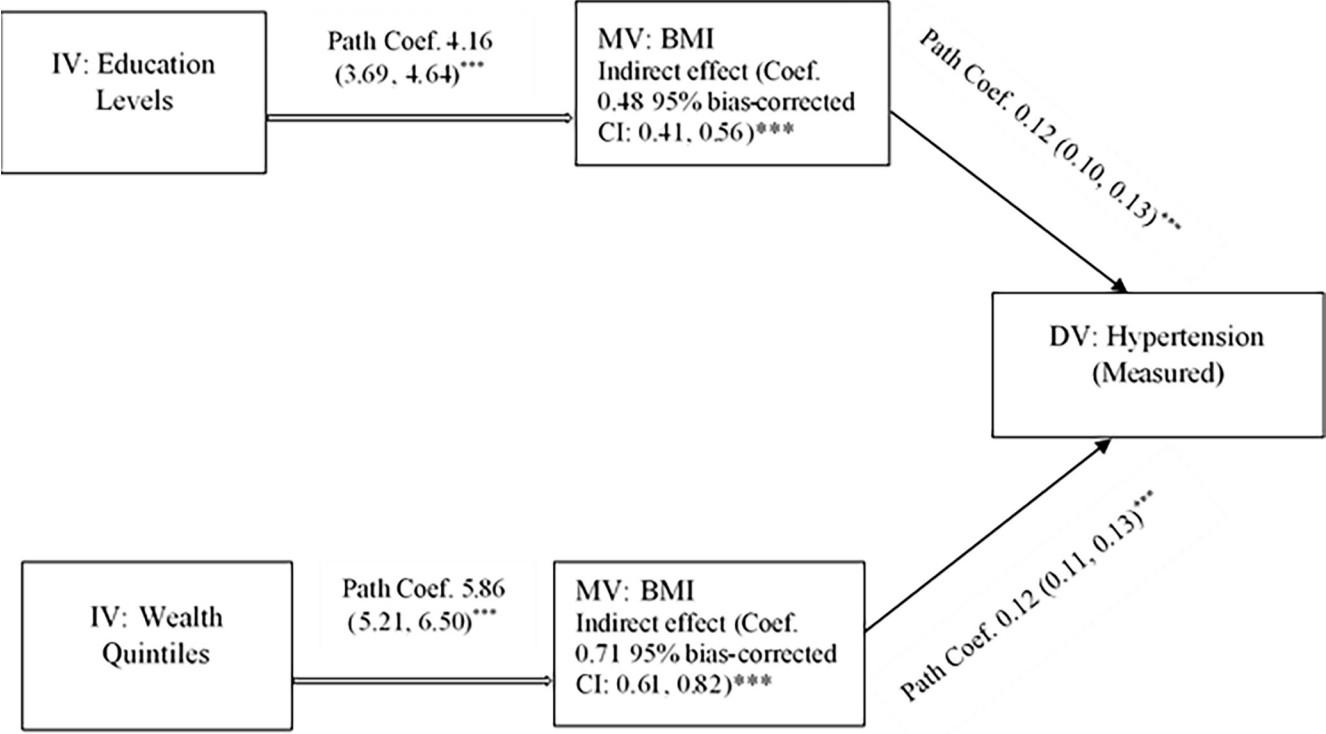

**Fig 3.** Mediating role of BMI in the association between SES and hypertension (measured) in Nepal. Path coefficients (95% CI) and indirect effect of SES on hypertension through BMI with bias-corrected 95% confidence intervals are reported. * $p<0.05$, ** $p<0.01$, *** $p<0.001$. CI = Confidence Interval, BMI = Body Mass Index, DV = Dependent Variable, IV = Independent Variable, MV = Mediating Variable, SES = Socioeconomic Status.

### Sensitivity analyses

We conducted sensitivity analyses that assessed the associations between SES and hypertension (measured and medical) adjusted for potential confounders according to the new guidelines of ACC/AHA 2017, which were stratified by sex and urbanity (S2 File). These analyses produced estimates and trends that are very similar to those for primary analyses, which reinforce our findings that increasing SES is associated with an increased likelihood of having hypertension, which modified by sex and urbanity.

Our sensitivity analyses constructed nested logistic regression models for the associations between hypertension (measured and medical) and SES that progressively adjusted for age, sex, urbanity, marital status, exposure to second-hand smoke, and BMI (Table 4, S1 File). The estimates and trends reinforce our primary findings.

For the secondary outcome, we examined the associations between SES and BMI using two different approaches. Firstly, we conducted sex and urbanity stratified analyses for SES using both global and South Asia specific categories of BMI (S1 and S2 Files). Secondly, we tested BMI as a continuous variable in association with SES due to the low prevalence of obesity (S1 File). These results supported that the likelihood of being overweight/obese increased with an increasing level of SES, which also modified by sex and urbanity.

We also conducted mediation analysis for hypertension by new guidelines of ACC/AHA 2017, which reinforce our argument that the association between SES and hypertension is mediated by BMI (S2 File).

## Discussion

Our study, including 13,436 people from a nationally representative survey, finds that people with increasing levels of SES (education and wealth) are at an increased risk of having hypertension in Nepal, with the association (education) moderated by gender. These associations also modified by urbanity. Our novel finding is that BMI mediated the associations between SES and hypertension in the context of LMICs, particularly in Nepal. We found these results were comparable for both the JNC7 and the ACC/AHA 2017 guidelines.

Established evidence suggests that risk factors for cardiovascular disease (CVD), including hypertension, are highly prevalent in low SES groups in developed countries [12, 26]. In contrast to this evidence, our study shows that the prevalence of hypertension was greater among people with higher SES groups, which is consistent with recent studies conducted in LMICs, particularly in South Asia [15, 27–30]. Substantial differences between men and women were observed only in the association between education and hypertension, which is consistent with previous studies in developed countries [15, 31–34]. However, a recent study claimed that the evidence of CVD risk among higher SES group in low-income countries is limited to particular countries and argued that the risk of CVD in low-income countries is higher among people with lower levels of education [35]. The study, however, did not investigate whether the risk of hypertension would be the same as the risk of CVD that warrant further research [35]. Moreover, we believe that this argument is merely applicable in the south Asian settings particularly in Nepal, due to recent economic and demographic transition [7, 15, 21].

In line with these studies, our study observed that increasing levels of education and wealth quintiles have a positive association with higher likelihood of BMI both in men and women. Hence, we formally tested the mediating roles of BMI in the association between higher SES and hypertension and demonstrated that BMI attenuates the observed associations. In other words, BMI may help to explain broader SES differentials in hypertension, particularly by education and wealth quintiles. Evidence from higher-income countries also supported that BMI mediates the association between education and the risk of cardiovascular diseases [36].

Our observed results have several policy implications. The comprehensive understanding of the mechanisms of socioeconomic differentials in hypertension may help to take adequate measures for the prevention of risk of CVD in resource-poor settings. Findings related to SES by sex and rural-urban differences in hypertension will also guide to take gender-sensitive policy measures in reducing CVD and its modifiable risk factors.

Our study demonstrated that the prevalence of obesity/overweight, according to the South Asia-specific cut-offs of BMI, was at unhealthy levels, and the risk of being obese/overweight was increased by the increasing levels of SES (education and wealth)." Thus, the identification of BMI as a mediator of the higher SES and hypertension association emphasizes on this modifiable risk factor as a potential target for interventions to reduce CVD and related risk factors such as hypertension and elevated blood pressure in higher SES groups in LMICs. This study provides further evidence allied to the emergence of SES gradients in CVD and related risk factors. Although few recent studies found SES gradients in CVD risk in LMICs setting, this research contributes to previous work by bridging the fields of socioeconomic differentials in CVD risk and formally testing established theoretical models. The veracity of our findings is contingent on replication with longitudinal data and more comprehensive assessments of SES.

To the best of our knowledge, this is the first study found mediating roles of the modifiable risk factors of CVD in the SES and hypertension association using a nationally representative sample in a resource-poor setting. Our study also first time assessed the association between SES and hypertension according to standard hypertension JNC7 guideline and a new guideline recommended by the ACC/ AHA 2017. We observed sex and rural-urban differences in blood

pressure outcomes by sex and urbanity stratified analysis. For instance, recent studies also emphasized to investigate the SES gradient along with sex and rural-urban differences in blood pressure outcomes in Nepal [7, 8, 11, 37].

Along with these novel contributions and methodological strengths, some limitations may also be considered with the interpretation of the results. We were not able to assess the causality of the associations between SES and hypertension due to the cross-sectional nature of the data. Our measurement of SES omits an indicator of employment status, which should be assessed in detail in further research. Finally, blood pressure measurement error may occur due to the quality of medical staff training in various regions of Nepal, even though an automatic device for BP measurement had been used.

## Conclusions

In conclusion, higher SES was positively associated with the higher likelihood of having hypertension in Nepal according to both JNC7 and ACC/AHA 2017 guidelines. All of the observed trends were more pronounced in men than in women, and there was evidence of differences in these trends between residents in rural and urban areas. The association between higher SES and hypertension was mediated by BMI, which may help to explain broader socioeconomic differentials in CVD and related risk factors, particularly in terms of education and wealth. Our study suggests that the mediating factor of BMI should be tackled to diminish the risk of CVD in people with higher SES in LMICs.

## Supporting information

**S1 File. Supporting tables.**
(DOCX)

**S2 File. Supporting figures.**
(DOCX)

## Acknowledgments

The authors thank to MEASURE DHS for granting access to the Nepal Demographic and Health Survey 2016 data. We also thank Prof. Parisa Tehranifar for her critical comments and suggestions on the initial version.

## Author Contributions

**Conceptualization:** Juwel Rana, Rakibul M. Islam.

**Data curation:** Juwel Rana.

**Formal analysis:** Juwel Rana, Kanchan Kumar Sen, Rakibul M. Islam.

**Methodology:** Kanchan Kumar Sen.

**Writing – original draft:** Juwel Rana, Zobayer Ahmmad, Kanchan Kumar Sen, Sanjeev Bista, Rakibul M. Islam.

**Writing – review & editing:** Zobayer Ahmmad, Kanchan Kumar Sen, Rakibul M. Islam.

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
