## [Decision Letter · Decision Letter 0]

12 Aug 2019

PONE-D-19-16119 Socioeconomic Differentials in Hypertension based on JNC7 and ACC/AHA 2017 Guidelines Mediated by Body Mass Index: Evidence from Nepal Demographic and Health Survey PLOS ONE Dear Mr Rana, Thank you for submitting your manuscript to PLOS ONE. After careful consideration, we feel that it has merit but does not fully meet PLOS ONE’s publication criteria as it currently stands. Therefore, we invite you to submit a revised version of the manuscript that addresses the points raised during the review process.

Based on reviewer comments and my read of your paper, I believe there are several areas where your paper can be strengthen.  First, you need streamline your paper to engage Reviewer 2’s concerns about overlapping confidence intervals and the conclusions drawn for your two measures.  Depending on the outcome (hypertension or blood pressure), the within and between group confidence intervals overlap considerably across education and wealth for men and women.  What does this tell us about gender and health in Nepal?  Furthermore, what explains the within group divergence for a particular level of socioeconomic status by gender?

Second, more information is needed about your mediation methods.  Reviewer 2 draws attention to how your analysis diverges from usual methods of mediation and that Figure 3 is missing from the manuscript.  I concur and would recommend that you conduct a more standard mediation model where the direct and indirect effects are estimated, with BMI as the mediator, for each gender and reported in a table or figure. 

We would appreciate receiving your revised manuscript by Sep 26 2019 11:59PM. To enhance the reproducibility of your results, we recommend that if applicable you deposit your laboratory protocols in protocols.io, where a protocol can be assigned its own identifier (DOI) such that it can be cited independently in the future. For instructions see: http://journals.plos.org/plosone/s/submission-guidelines#loc-laboratory-protocols

We look forward to receiving your revised manuscript.

Kind regards,

Bryan L. Sykes, Ph.D.

Academic Editor

PLOS ONE

2. Please upload a copy of Figure 3, to which you refer in your text on page 18. If the figure is no longer to be included as part of the submission please remove all reference to it within the text.

Reviewers' comments:

Reviewer's Responses to Questions

**Comments to the Author**

1. Is the manuscript technically sound, and do the data support the conclusions?

Reviewer #1: Yes

Reviewer #2: No

2. Has the statistical analysis been performed appropriately and rigorously? 

Reviewer #1: Yes

Reviewer #2: No

3. Have the authors made all data underlying the findings in their manuscript fully available?

Reviewer #1: Yes

Reviewer #2: No

4. Is the manuscript presented in an intelligible fashion and written in standard English?

Reviewer #1: Yes

Reviewer #2: Yes

5. Review Comments to the Author

Reviewer #1: The authors have written a very interesting manuscript. The aim of the manuscript is to assess the associations between SES

and hypertension in Nepal and the extent to which these associations vary by sex and urbanity. The authors used a robust methodology to estimate the mediating role of body mass index. I do not have much comments about the results, however, I have a few minor comments and only one major concern.

1. The authors could fortify their results by providing the empirical strategy they used for the estimation. This is very important for other researchers who would like replicate their work.

2. I think the authors could recommend some policy implications of their results.

3. The figures in the manuscript are blare.

Reviewer #2: The methodology used by the authors seems to be commonly used for this type of analysis but it is not properly used and described. The results that they report contain important flaws from a methodological point of view. I explain my concerns with more detail in the attached file.

6. PLOS authors have the option to publish the peer review history of their article (what does this mean?). If published, this will include your full peer review and any attached files.

Reviewer #1: No

Reviewer #2: No

---

## [Author Response · Author response to Decision Letter 0]

27 Oct 2019

26th October 2019

Bryan L. Sykes, Ph.D.

Academic Editor

PLOS ONE

RE: PONE-D-19-16119 ‘Socioeconomic differentials in hypertension based on JNC7 and ACC/AHA 2017 guidelines mediated by body mass index: Evidence from Nepal demographic and health survey.’

Dear Professor Sykes,

We thank you for the opportunity to resubmit our manuscript and for the considered comments of the reviewers. We have addressed each of these point by point below, and a revised version of the manuscript is submitted for your consideration.

We believe we have addressed all the reviewers’ comments, and where changes have been made, have shown these as marked yellow changes. We hope the manuscript is now considered acceptable for publication in your journal.

Yours sincerely,

Juwel Rana, MPH 

Corresponding author 

Editor’s Concern

Based on reviewer comments and my read of your paper, I believe there are several areas where your paper can be strengthen. First, you need streamline your paper to engage Reviewer 2’s concerns about overlapping confidence intervals and the conclusions drawn for your two measures. Depending on the outcome (hypertension or blood pressure), the within and between group confidence intervals overlap considerably across education and wealth for men and women. What does this tell us about gender and health in Nepal? Furthermore, what explains the within group divergence for a particular level of socioeconomic status by gender?

Authors’ Response: We appreciate your succinct summarization of reviewers’ comments. We have addressed all of your concerns regarding the manuscript. 

Second, more information is needed about your mediation methods. Reviewer 2 draws attention to how your analysis diverges from usual methods of mediation and that Figure 3 is missing from the manuscript. I concur and would recommend that you conduct a more standard mediation model where the direct and indirect effects are estimated, with BMI as the mediator, for each gender and reported in a table or figure.

Authors’ Response: We apologize that the missing of figure 3 about mediation analysis introduced confusion of reviewer 2. We have now provided the missing figure 3, indirect, and direct effects estimate of SES with details of the mediation analysis process. 

Authors’ Response: Thank you for your guidance. We have formatted the title, headings, subheadings, and authors’ affiliation according to the journal requirements. 

2. Please upload a copy of Figure 3, to which you refer in your text on page 18. If the figure is no longer to be included as part of the submission, please remove all reference to it within the text.

Authors’ Response: We apologize for this unintentional error, which raised some questions related to mediation analysis. We believe the figure will make more sense about the mediation analysis. We have now uploaded the Figure 3. 

Reviewer #1: The authors have written a very interesting manuscript. The aim of the manuscript is to assess the associations between SES and hypertension in Nepal and the extent to which these associations vary by sex and urbanity. The authors used a robust methodology to estimate the mediating role of body mass index. I do not have much comments about the results, however, I have a few minor comments and only one major concern.

1. The authors could fortify their results by providing the empirical strategy they used for the estimation. This is very important for other researchers who would like replicate their work.

Authors’ Response: Thank you very much for your appreciation. We have now included more details on the methods and material section along with the particular references for the empirical strategy that we followed. 

2. I think the authors could recommend some policy implications of their results.

Authors’ Response: We have already discussed some policy implications of our results on Page 20-21. 

3. The figures in the manuscript are blare.

Authors’ Response: The quality of figures was distorted due to the conversion of JPG to TIFF. We have now replaced those with a clear version of the figures.

Reviewer #2: 

The main objective of this research is to evaluate if there is a positive association between socioeconomic status (SES) and the probability of having hypertension in Nepal. They use the 2016 Nepal Demographic and Health Survey to estimate logit models using binary dependent variable based on two definitions of blood pressure. One dependent variable is only based on the blood pressure exams taken during the interview. The second dependent variable identifies high-blood pressure individuals based on the blood pressure exams taken during the interview or on self-reported diagnosis of high pressure or medicine prescription. In addition, the authors include in the models an obesity-related variable to evaluate if the body mass index could be viewed as a mediator variable between SES and the likelihood of having high blood pressure. The authors conclude that higher SES was positively associated with the higher likelihood of having hypertension, and that such association between higher SES and hypertension was mediated by the body mass index (BMI). 

The methodology used by the authors seems to be commonly used for this type of analysis, but it is not properly used and described. More importantly, conclusions do not agree with the statistical result. I explain below my concerns with more detail.

Main Comments

1. The authors conclude that higher SES is positively associated with a higher likelihood of having hypertension. However, this conclusion is not reliable because of two reasons: 

(a) Most of the confidence intervals (CIs) of the odds ratios (ORs) related to education or wealth overlap with each other. In particular, the confidence interval for the highest level of SES (education or wealth) is the widest. Therefore, it is possible that ORs are not significantly different across SES levels. If there is some significant difference based on the results, it would be for any SES with respect to the lowest SES where in which it is more likely to observe under-nutrition (and probably a very low prevalence rate of obesity). These are just two examples to illustrate extreme CI overlap: (i) S1 Table, medical, women: primary education CI [1.24, 1.94], higher education CI [1.18, 2.25] (ii) S2 Table, medical, overall: poorer CI [1.08, 1.52], richer CI [1.02, 1.59].

Authors’ Response (a): Thank you for raising this point. Firstly, we have now presented the results of the associations between SES and hypertension (measurement-only definition) addressing the reviewer’s concern. Our comparators were ‘no education/preschool’ (Ref. Category) for other levels of education and ‘poorest wealth quintile’ (Ref. Category) for other categories of wealth quintiles, which indicate that we do not need to be cautious about the overlap of CI across SES levels. Therefore, our conclusion about positive associations between higher SES and hypertension is reliable. 

Both Editor and reviewer’s concern about the between-group confidence intervals overlap across SES is more reasonable to investigate further because we did not formally test effect modification/interaction by sex in the earlier version. However, we have now tested effect modification of sex by interaction term along with previous sex-stratified analysis, which clarifies overlaps of confidence intervals across SES levels between men and women. Our effect modification test confirms that the effect of higher levels of education on hypertension was different between men and women, but the effect of wealth on hypertension was not different between men and women. Therefore, we do not have methodological limitations to conclude that higher SES is positively associated with a higher likelihood of having hypertension, which is different by sex across education but not across wealth quintiles.

We also would like to note that there has been an entrenched belief, especially in medicine, that overlapping 95%CIs is statistical insigniﬁcance, p>0.05. However, there have now been a number of explanations that prove that this belief is incorrect. If two statistics have non-overlapping confidence intervals, they are necessarily significantly different, but if they have overlapping confidence intervals, it is not necessarily true that they are not significantly different. 

Several writers have also pointed out that considerable overlap can be compatible with a signiﬁcant difference, p=0.05 (i-v). 

i. Andrea Knezevic A. Overlapping Confidence Intervals and Statistical Significance. The Cornell Statistical Consulting Unit, StatNews # 73, 2008. 

ii. Cumming G, Finch S. Inference by eye: confidence intervals and how to read pictures of data. American Psychologist 2005; 60:170–180. DOI: 10.1037/0003-066X.60.2.170.

iii. Schenker N, Gentleman JF. On judging the signiﬁcance of differences by examining the overlap between conﬁdence intervals. The American Statistician 2001; 55:182–186. DOI: 10.1198/000313001317097960. 

iv. Austin PC, Hux JE. A brief note on overlapping conﬁdence intervals. Journal of Vascular Surgery 2002; 36:194–195. DOI: 10.1067/mva.2002.125015. 

v. Cumming G. Inference by eye: Reading the overlap of independent confidence intervals. Statist. Med. 2009; 28:205–220

(b) It is not clear how the medical high blood pressure is defined. It seems that adults that do not have high pressure based on the blood pressure readings could still be considered to have high pressure if they reported a past diagnosis of hypertension or they report that are taking some prescribed medicine for high blood pressure. Since access to health services could be correlated with higher SES, it is possible that this medical definition of high blood pressure biases the estimated relationship between SES and the probability of having high blood pressure. In fact, OR point estimates using the medical definition of high blood pressure are systematically larger than those calculated using the measured blood pressure definition, whereas their corresponding CI are always to the right of those calculated with the measured blood pressure (e.g., see Table Sl-S4). This limitation of using the medical definition should be clearly stated in the manuscript.

Authors’ Response (b): We appreciate reviewer’s concern regarding the medical definition of hypertension used in this study based on published literature. Following the reviewer’s suggestions, we have now reassessed the association between SES and hypertension based on ‘measurement-only’ definition and presented throughout the manuscript to avoid biases and limitations of using medical definition suspected by the reviewer. 

2. Given the limitation of the medical definition of high blood pressure, the exercise of mediation presented in Table 4 should be done with the measurement-only definition. It is not clear why the authors only present the results using the medical definition. In addition, Table 4 presents results as exponentiated coefficients whereas the OR estimates are reported for other models. I suggest to present the results of Table 4 using the OR estimates. Since other regressors are included exponentiated coefficients are not equal to the OR.

Authors’ Response: According to the reviewer’s recommendation, we have performed a mediation analysis with the ‘measurement-only’ definition of hypertension. Table 4 presents regression coefficients (exponentiated coefficients) because we can initially observe the mediation effect of BMI on the basis of a rule thumb-comparing at least 10% change in coefficients for models with and without adjusting for the mediator variable. We cannot calculate/compare a 10% change in terms of OR that would produce misleading results. Thus, we had to report regression coefficients rather than OR. 

3. The mediator analysis is poorly explained. The authors say: "The second step was the "indirect effect "approach, which formally examined the statistical significance of an indirect effect using the product of coefficients approach. For assessing the indirect effect of BMI on these binary outcomes, we used the binary-mediation package in Stata" The results of this analysis are poorly reported (e.g., Figure 3 is not in the manuscript). Furthermore, to the best of my knowledge, there are at least three different methods in Stata to conduct mediation analysis. I suggest that the authors explain this procedure clearly.

Authors’ Response: We apologize for this unintentional error, as Figure 3 was not uploaded in the previous version. Following the editor and reviewer’s suggestions, we have now reported the results (Figure 3 is attached) of direct and indirect effect with bias-corrected confidence intervals. We acknowledge that there are multiple methods of mediation analysis. However, we used generalized structural equation modeling (GSEM) technique. We are happy to upload STATA code as supplementary material if the editor asks for. The procedure is explained in detail on the materials and methods section in our manuscript as well as in the following articles:

i. https://stats.idre.ucla.edu/stata/faq/how-can-i-do-mediation-analysis-with-the-sem-command/

ii. https://stats.idre.ucla.edu/stata/faq/how-can-i-do-mediation-analysis-with-a-categorical-iv-in-stata/

iii. StataCorp. 2019. Stata: Release 16. Structural Equation Modeling Reference Manual, Example 42

iv. Baron RM, Kenny DA. The Moderator-Mediator Variable Distinction in Social Psychological Research. Conceptual, Strategic, and Statistical Considerations. J Pers Soc Psychol. 1986;51: 1173–82. doi:10.1037/0022-3514.51.6.1173

v. Preacher KJ, Hayes AF. Asymptotic and resampling strategies for assessing and comparing indirect effects in multiple mediator models. Behav Res Methods. 2008;40: 879–891. doi:10.3758/BRM.40.3.879 

vi. Hosmer DW, Lemeshow S, Sturdivant RX. Applied Logistic Regression: Third Edition. Applied Logistic Regression: Third Edition. 2013. doi:10.1002/9781118548387

4. According to Nepal DHS, 2016-Final Report, rates of hypertension are higher among tobacco users. Therefore, the mediation analysis should include this variable. 

Authors’ Response: This is a good point raised by the reviewer, and we have already considered secondhand smoking in the mediation analysis. There are several reasons for using secondhand smoking over tobacco users. First, since tobacco can be used in several forms (cigarette smoking, chewing with battle leaf and water pipes, also referred to as Hookah, Shisha, Narghile, Argileh) in South Asian countries, including Nepal, comprehensive data on ‘tobacco use’ was unavailable. Secondly, the small sample size of current tobacco smoking that will reduce the power of the analysis. 

5. According to the Nepal DHS, 2016-Final Report l "The average of the second and third measurements was used to classify the respondent with respect to hypertension, according to internationally recommended categories (WHO 1999; NIH 1997) l'. It is not clear why the authors use the average of the three measurements. Would the results be different if they use the variable created and suggested by the Nepal DHS? See Tables 14.3.1 and 14.32 to find the definitions of hypertension available in Nepal DHS. 2016.

Authors’ Response: Our research aimed to compare between JNC7 and ACC/AHA 2017 guidelines. Thus, we have classified hypertension according to the internationally recommended categories of JNC7 and ACC/AHA 2017, which recommended to take the average of the two or more measurements for reducing the bias of measurement error. The previously published literature also used the average of three measurements for ensuring higher accuracy (Harshfield E, Chowdhury R, Harhay MN, Bergquist H, Harhay MO. Association of hypertension and hyperglycaemia with socioeconomic contexts in resource-poor settings: The Bangladesh Demographic and Health Survey. Int J Epidemiol. 2015; 44: 1625–1636. doi:10.1093/ije/dyv087). 

According to the reviewer’s suggestion, using the average of the two (second and third) measurements, the results (15.6% women and 22.1% men) were not much different from the Nepal DHS (17% women and 23% men). The existing variation of prevalence could be explained by the differences in age groups included in the analyses. For example, the Nepal DHS included men and women aged 15+ while we included men and women aged 18+ in our analysis. 

Furthermore, our results (16% women and 22.8% men aged 18 years and older) based on the average of three measurements are almost identical to the Nepal DHS report (17% women and 23% men aged 15 and older). 

6. It is not clear if sampling weights were included in the estimations of the logistic regressions. They should be used.

Authors’ Response: Thank you. Yes, we have used sampling weights, which is already mentioned on page 6, lines 125-127 and page 10, lines 211-213, and in Supplementary Fig 1. 

7. About policy implications. The authors suggest that interventions should be done in the higher SES groups to modify hypertension or obesity. Based on my previous comments, the results that they report contain important flaws from a methodological point of view, so they do not support this policy recommendation.

Authors’ Response: Thank you for your critical remarks. We have already considered reviewer’s suggestion and presented results by measurement-only definition which give an almost identical conclusion. Moreover, the Nepal DHS Report 2016 also gave a similar conclusion based on their results (Nepal DHS Figure 14.3) that is identical to our findings. However, we are now cautious about our argument in the discussion, Page 19. 

Minor Comments

1. The number of observations mentioned by the authors is smaller from the one reported in the Nepal DHS, 2016-Final Report (see Tables 14.3.1 and 14.3.2). Is it possible that the range of ages considered was only between 15-60 years old? In the case of BMI the number of observations reported in Nepal DHS seems to be different too (e.g. it was not measured for pregnant women). I suggest that the authors report the sample size for each estimated model.

Authors’ Response: We reported now the sample size for the all estimated models. However, we had mentioned in the previous version about the range of age (18 years and older) and the number of unweighted and weighted observations in Tables 2 and 3, Page 6 Lines 125-127; Figures 1 and 2, including in the Supplementary Fig 1. 

2. The evidence that prevalence of hypertension is higher among high SES in low income countries is limited to specific countries and it has been questioned. Furthermore, in a recent study Rosengrent et al. (2019) conclude that "people with low levels of education in low-income and middle-income countries had a markedly higher risk of major cardiovascular events compared with those with higher levels of education. Cardiovascular disease in low-income countries is a problem predominantly among people with lower levels of education, whereas the situation in middle-income countries is more variable" I suggest that the authors consider this issue at least in the discussion section.

Authors’ Response: Thank you for this point and suggesting relevant literature. We already used the referred article in the earlier version (Ref. 28 Razak F and Subramanian S. Commentary: Socioeconomic status and hypertension in low- and middle-income countries: can we learn anything from existing studies? Int J Epidemiol 2014; 43(5): 1577-1581). We further acknowledge reviewer’s concern and added the following text on page 19, lines 356-361 to be cautious about our conclusion.

“However, a recent study claimed that the evidence of CVD risk among higher SES groups in low-income countries is limited to particular countries and argued that the risk of CVD in low-income countries is higher among people with lower levels of education.35 The study, however, did not investigate whether the risk of hypertension would be the same as the risk of CVD that warrant further research.35 Moreover, we believe that this argument is merely applicable in the south Asian settings particularly in Nepal, due to recent economic and demographic transition.7, 15, 21”

3. Improve quality of figures.

Authors’ Response: We acknowledge the poor quality of figures due to the conversion of JPG to TIFF. However, we have now replaced those with a clear version of the figures.

---

## [Decision Letter · Decision Letter 1]

18 Dec 2019

PONE-D-19-16119R1

Socioeconomic Differentials in Hypertension based on JNC7 and ACC/AHA 2017 Guidelines Mediated by Body Mass Index: Evidence from Nepal Demographic and Health Survey

PLOS ONE

Dear Mr Rana,

Thank you for submitting your manuscript to PLOS ONE. After careful consideration, we feel that it has merit but does not fully meet PLOS ONE’s publication criteria as it currently stands. Therefore, we invite you to submit a revised version of the manuscript that addresses the points raised during the review process.

This manuscript has improved greatly.  Thank you for taking seriously the reviewer comments and my suggestions from the last round of review.  There are several outstanding revisions in need of elaboration or correction before your paper can be accepted for publication.  Should you make the following set of revisions successfully, your revised submission will not be sent out for re-review, and I will accept your paper for publication upon resubmission.

First, Reviewer 2 draws your attention to an overarching conclusion espoused in the paper.  I concur with the reviewer’s assessment and request that you change the statement **from** “Our study, including 13,436 people from a nationally representative survey, demonstrated that increasing levels of SES (education and wealth) were positively associated with an increased risk of having hypertension in Nepal, with an evidence of effect modification of gender in the 351 association between education and hypertension” **to** “Our study, including 13,436 people from a nationally representative survey, finds that higher levels of education (compared to individuals with no education or at a pre-school educational level) are at an increased risk of having hypertension in Nepal, with the association moderated by gender.”

Second, Reviewer 2 raises a question that merits engagement in your discussion section.  Namely, how does BMI mediate hypertension in a population that has normal (or healthy) BMI levels?  

We would appreciate receiving your revised manuscript by Feb 01 2020 11:59PM. To enhance the reproducibility of your results, we recommend that if applicable you deposit your laboratory protocols in protocols.io, where a protocol can be assigned its own identifier (DOI) such that it can be cited independently in the future. For instructions see: http://journals.plos.org/plosone/s/submission-guidelines#loc-laboratory-protocols

We look forward to receiving your revised manuscript.

Kind regards,

Bryan L. Sykes, Ph.D.

Academic Editor

PLOS ONE

Reviewers' comments:

Reviewer's Responses to Questions

**Comments to the Author**

1. If the authors have adequately addressed your comments raised in a previous round of review and you feel that this manuscript is now acceptable for publication, you may indicate that here to bypass the “Comments to the Author” section, enter your conflict of interest statement in the “Confidential to Editor” section, and submit your "Accept" recommendation.

Reviewer #1: All comments have been addressed

Reviewer #2: (No Response)

2. Is the manuscript technically sound, and do the data support the conclusions?

Reviewer #1: Yes

Reviewer #2: Partly

3. Has the statistical analysis been performed appropriately and rigorously? 

Reviewer #1: Yes

Reviewer #2: (No Response)

4. Have the authors made all data underlying the findings in their manuscript fully available?

Reviewer #1: No

Reviewer #2: (No Response)

5. Is the manuscript presented in an intelligible fashion and written in standard English?

Reviewer #1: Yes

Reviewer #2: Yes

6. Review Comments to the Author

Reviewer #1: Specific comments

Page 11: line 232 - 233; I think the reviewer should be cautious in interpreting their values since they claim there were too many missing values.

Table 2: sample characteristics. I suggest that the authors interpret the dummies from averages to frequencies.

Page 29: line 389 - 396 should go to the conclusions.

General

Your conclusion should address the summary, limitations and policy recommendations. Please do restructure your introduction.

Reviewer #2: (No Response)

7. PLOS authors have the option to publish the peer review history of their article (what does this mean?). If published, this will include your full peer review and any attached files.

Reviewer #1: No

Reviewer #2: No

---

## [Author Response · Author response to Decision Letter 1]

20 Dec 2019

20th December 2019

Bryan L. Sykes, Ph.D.

Academic Editor

PLOS ONE

RE: PONE-D-19-16119 ‘Socioeconomic differentials in hypertension based on JNC7 and ACC/AHA 2017 guidelines mediated by body mass index: Evidence from Nepal demographic and health survey.’

Dear Professor Sykes,

We thank you for the opportunity to resubmit our manuscript and for the considered comments of the reviewers. We have addressed each of these point by point below, and a revised version of the manuscript is submitted for your consideration.

We believe we have addressed all the reviewers’ comments, and where changes have been made, have shown these as marked yellow changes. We hope the manuscript is now considered acceptable for publication in your journal.

Yours sincerely,

Juwel Rana, MPH (double)

Corresponding author 

Editor’s Concern

First, Reviewer 2 draws your attention to an overarching conclusion espoused in the paper. I concur with the reviewer’s assessment and request that you change the statement from “Our study, including 13,436 people from a nationally representative survey, demonstrated that increasing levels of SES (education and wealth) were positively associated with an increased risk of having hypertension in Nepal, with an evidence of effect modification of gender in the 351 association between education and hypertension” to “Our study, including 13,436 people from a nationally representative survey, finds that higher levels of education (compared to individuals with no education or at a pre-school educational level) are at an increased risk of having hypertension in Nepal, with the association moderated by gender.”

Authors’ Response: We appreciate your succinct summarization of reviewers’ comments. Reviewer 2’s concern is about overlapping CIs in the associations. In our first round responses, we benignly addressed the issues of CI’s overlapping. However, the concept of overlapping CI’s is only relevant when the outcome measure is continuous and normally distributed. Our outcome is binary, and effect estimates are odds ratios, which are NOT normally distributed. Like all other non-normal estimates such as risk ratio or hazard ratio, the issues of overlapping CIs cannot be applied to the odds ratio. The lower limits and upper limits of odds ratio are exponential of the margin of error of beta coefficients. Hence, the conclusion drawn from the results is valid.

Moreover, despite gender-stratified analysis, we have formally tested effect modification of gender in the associations between SES and hypertension. It reaffirms our rationale related to CI’s overlapping. Therefore, we would like to keep the statement as it currently appears in the manuscript. 

However, if the editor feels the statement needs to be changed, we are happy to change it, but the statement will be partially correct based on our results. 

According to editor’s suggestions, we have modified the tone of our statement in page 19, lines 352-354: “Our study, including 13,436 people from a nationally representative survey, finds that people with increasing levels of SES (education and wealth) are at an increased risk of having hypertension in Nepal, with the association (education) moderated by gender.”

Second, Reviewer 2 raises a question that merits engagement in your discussion section. Namely, how does BMI mediate hypertension in a population that has normal (or healthy) BMI levels? 

Authors’ Response: We agree with this point. Due to getting low prevalence of obesity, according to the global cut-offs of BMI, we counted and categorized BMI as well as obesity/overweight according to the South Asia-specific cut-offs for BMI in our earlier version and already showed on page 19, lines 341-346 and in supplementary tables 5 and 6. 

However, we have provided further clarification in the result and discussion section now according to the suggestions of the editor and reviewer in the following pages and lines:

Result: Page 17, lines 306-310: “The average BMI, according to the global cut-offs of BMI, was about 22, which indicates about 18% of the respondents were obese/overweight. However, the prevalence of obesity/overweight, according to the South Asia-specific cut-offs of BMI, was about 37%. The likelihood of being overweight/obese increased with an increasing level of SES, which also modified by sex and urbanity (S5 Table, S6 Fig).”

Discussion: Page 21, lines 385-387: “Our study demonstrated that the prevalence of obesity/overweight, according to the South Asia-specific cut-offs of BMI, was at unhealthy levels, and the risk of being obese/overweight was increased by the increasing levels of SES (education and wealth).”

Reviewer #1: Specific comments

Page 11: line 232 - 233; I think the reviewer should be cautious in interpreting their values since they claim there were too many missing values.

Authors’ Response: We completely agree with this point. This was one of the reasons not considering employment as an indicator of SES in our study. Thus, we are cautious about our interpretation and we did not include ‘employment’ as one of the main independent variables. 

Table 2: sample characteristics. I suggest that the authors interpret the dummies from averages to frequencies.

Page 29: line 389 - 396 should go to the conclusions.

General

Your conclusion should address the summary, limitations and policy recommendations. Please do restructure your introduction.

Authors’ Response: We think now we have included more details, which addressed these comments. Moreover, our conclusion seems inclusive addressing summary, limitations and policy recommendations. 

Reviewer #2: 

A) I appreciate the effort of the authors to respond to all my comments. In my opinion, the quality and the clarity of the paper have improved significantly. However, I have one concern about this conclusion: 

“Our study, including 13,436 people from a nationally representative survey, demonstrated that increasing levels of SES (education and wealth) were positively associated with an increased risk of having hypertension in Nepal, with an evidence of effect modification of gender in the 351 association between education and hypertension”. 

I disagree with this conclusion because the statistical results that they obtain do not constitute a demonstration or a proof that increasing levels of SES are positively associated with hypertension in Nepal. 

In fact, the statistical results present evidence that adults with some level of formal education (primary or beyond) are more likely to have hypertension in Nepal. My conclusion is based on the statistical results of Figure 1 and 2 (and the corresponding estimated models from Table 4). These figures show that the confidence intervals (CIs) of the odd ratios (ORs) related to education or wealth overlap with each other. This is the same comment that I included in my previous review. The authors’ response was that “If two statistics have non‐overlapping confidence intervals, they are necessarily significantly different, but if they have overlapping confidence intervals, it is not necessarily true that they are not significantly different.” That is totally correct. Since there are overlapping CIs, the ORs could or could not be different. Therefore, we cannot conclude that we have demonstrated that the probability of having hypertension increases as the SES increases. 

This is equivalent to the result of not rejecting a null hypothesis. In that case, we would never claim that the null is true (or not true). The most precise way to solve this concern is by testing the equality of the parameters estimated in the models presented in Table 4 (without the mediator). A Wald test or a Lagrange multiplier test can be used to evaluate if the null that the parameters associated with different SES are different is rejected or not. Once again, a clear conclusion from this article is that having primary education or more increases the probability of having hypertension. As I mentioned in my previous review, this result could be explained if people with the lowest SES are also those with under‐nutrition levels. 

Authors’ Response: We have addressed this point and mentioned it in the editor’s concern. 

B) Finally, the average BMI in Nepal is around 22 with a small standard deviation. This suggests that the prevalence rate of obesity is small. How could the BMI be an important mediator for hypertension in individuals when most of them have BMI levels that are considered healthy levels? It would be worth considering this issue in the discussion section.

Authors’ Response: We have addressed this point and mentioned it in the editor’s concern.

---

## [Editor Report · Decision Letter 2]

3 Jan 2020

Socioeconomic Differentials in Hypertension based on JNC7 and ACC/AHA 2017 Guidelines Mediated by Body Mass Index: Evidence from Nepal Demographic and Health Survey

PONE-D-19-16119R2

Dear Dr. Rana,

We are pleased to inform you that your manuscript has been judged scientifically suitable for publication and will be formally accepted for publication once it complies with all outstanding technical requirements.

With kind regards,

Bryan L. Sykes, Ph.D.

Academic Editor

PLOS ONE
---

## [Editor Report · Acceptance letter]

17 Jan 2020

PONE-D-19-16119R2 

Socioeconomic Differentials in Hypertension based on JNC7 and ACC/AHA 2017 Guidelines Mediated by Body Mass Index: Evidence from Nepal Demographic and Health Survey 

Dear Dr. Rana:

I am pleased to inform you that your manuscript has been deemed suitable for publication in PLOS ONE. Congratulations! Your manuscript is now with our production department. 

With kind regards,

on behalf of

Dr. Bryan L. Sykes 

Academic Editor

PLOS ONE